

# Water remote sensing reflectance from radiative transfer simulations on a global scale of Inherent Optical Properties

Lena Kritten[1], Rene Preusker[1], Carsten Brockmann[2], Tonio Fincke[2], Sampsa Koponen[3], and Jürgen Fischer[1]

[1]Freie Universität Berlin, Berlin, Germany
[2]Brockmann Consult, Geesthacht, Germany
[3]Finnish Environment Institute, Helsinki, Finland

**Correspondence:** (lena.kritten@wew.fu-berlin.de)

**Abstract.** The remote-sensing reflectance ($R_{rs}$) is in someway an artificial unit, that is constructed in order to contain the spectral colour information of the water body, but to be hardly influenced by the atmosphere above. In ocean colour remote-sensing it is the measure to define the optical properties of the water/water constituents. $R_{rs}$ is the ratio of water-leaving radiance and down-welling irradiance. It is derived from top-of-atmosphere radiance/reflectance measurements through atmospheric

correction. A database with $R_{rs}$s from radiative transfer simulations is capable to serve as a forward model for the retrieval of water constituents.

For the present database the $R_{rs}$ is simulated in dependency of inherent optical properties (IOPs) representing pure water with different salinities and 5 water constituents (Chlorophyll-a-pigment, Detritus, CDOM (coloured dissolved organic matter), a 'big' and a 'small' scatterer) in a global range of concentrations. The interpolation points for each IOP were chosen in order

to reproduce the entire functional relationship between this particular IOP and the corresponding $R_{rs}$. The IOPs are varied independently.

The data is available for 9 solar, 9 viewing zenith and 25 azimuth angles. The spectral resolution of the data is 1nm, which allows the convolution to any ocean colour sensors' spectral response function. The data is produced with the radiative transfer code MOMO (Matrix Operator Model), which simulates the full radiative transfer in atmosphere and ocean. The code is hosted

at the institute of space sciences at Freie Universität Berlin and is not publicly available. The look-up table (LUT) is available at: https://doi.org/10.1594/WDCC/LUT_for_WDC_I (Kritten et al., 2017).

## 1 Introduction

Many Databases including satellite based or insitu measured $R_{rs}$ do exist (Nechad et al. (2015),Barker (2013)). During the DLR/BMWi founded project SIOCS (Sensor Independent Ocean Colour Service) a LUT for the retrieval of geophysical pa-

rameters from $R_{rs}$ over ocean and inland waters was generated from radiative transfer modelling (RTM). This document describes the model and the simulation settings and includes a short Validation. The spectral remote-sensing reflectance $R_{rs}$ is



defined as

$$Rrs(\theta, \phi, \lambda) = L_w(\text{in air}, \theta, \phi, \lambda)/E \downarrow (\text{in air}, \lambda)[\text{sr}^{-1}] \tag{1}$$

where $L_w$ is the water-leaving radiance (Ocean Optics) and "in air" indicates that $R_{\rm rs}$ is evaluated using the water-leaving radiance $L_w$ and $E_d$ in the air, just above the water surface. In the case of satellite ocean colour, the spectral $R_{\rm rs}$ determined

from top of-atmosphere radiance is the primary data product used for the generation of higher level products such as chlorophyll a concentration.

## 2   Simulations

### 2.1   The RTM MOMO

The simulations are performed using the vector version of MOMO (Fell and Fischer (2001), Hollstein and Fischer (2012)).

Here a horizontal homogeneous atmosphere and ocean consisting of layers with vertical uniform optical properties are assumed. The upward and downward directed light field is calculated at all inter layer boundaries and for all solar positions which are defined in Sect. 2.1.1. The azimuthal dependence of the light field is internally expressed as Fourier series and reconstructed at equidistant distributed azimuth angles (see also Sect.2.1.1). The model is operated by several input files which govern the height profile of atmosphere and ocean, the scatterers, the absorber and the atmosphere ocean interface (Sect. 2.2 and 2.3).

### 2.1.1   Spectral and angular resolution

The Simulations are performed in order to reproduce the channels of ocean colour instruments. Wavelength range 390-1020nm in 1nm steps is suitable for this.

The simulations are azimuthally resolved with 120 Fourier terms and carried out for 10 sun and viewing zenith angles: $0°$, $11.2622°$, $20.6204°$, $29.9022°$, $39.1616°$, $48.4115°$, $57.6566°$, $66.899°$, $76.1399°$

Since MOMO is a non-spherical model, which neglects the earth's curvature, the data is only valid for the first 8 sun and viewing zenith angles.

Further, the simulations are carried out for 25 azimuth angles: $0°$, $7.5°$, $15°$, $22.5°$, $30°$, $37.5°$, $45°$, $52.5°$, $60°$, $67.5°$, $75°$, $82.5°$, $90°$, $97.5°$, $105°$, $112.5°$, $120°$, $127.5°$, $135°$, $142.5°$, $150°$, $157.5°$, $165°$, $172.5°$, $180°$. The azimuth angle is here defined as the difference between the sun and the viewing azimuth.

### 2.2   Atmosphere

The model atmosphere consists of 6 layers. In addition to molecular Rayleigh scattering one aerosol scatterer is considered: The spectral micro-physical data for a maritime aerosol at 90% humidity is taken from a database (Hess et al., 1998). Using this as input the phase functions of the aerosol are calculated using Mie Theory (Mie, 1908). The aerosol is located at 2km altitude with an optical thickness of $\tau$=0.2.



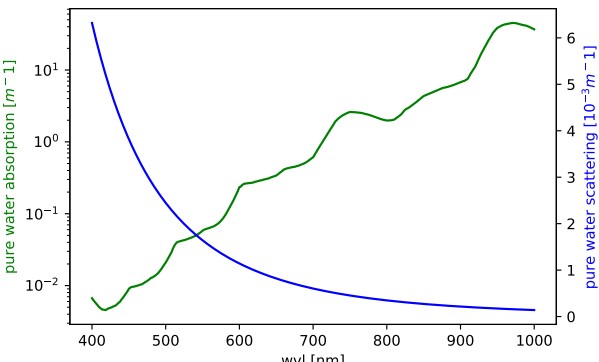

**Figure 1.** Spectral scattering (blue) and absorption (green) of pure seawater for salinity S=20 PSU and temperature T=20°C.

## 2.3 Water surface and water body

The wind blown ocean surface is described by wave facets whose normals are statistically distributed using the Cox and Munk model (Cox and Munk, 1954). The wind speed therein was set to 7 m/s.

The water body is assumed to be well mixed and 100m deep. In order to guarantee numerical stability, for high scattering cases we applied a dynamical layering. (For IOP definition see Sect. 2.3.2):

- if total back-scattering at 550nm > 10: water body depth=10m

- if total back-scattering at 550nm > 100: water body depth=1m

- if total back-scattering at 550nm > 400: water body depth=0.1m

### 2.3.1 Pure water absorption and scattering

The absorption coefficient of pure sea water (see Figure 1, right panel) is a result from the ESA project WATERRADIANCE (Röttgers et al., 2010) as a linear expansion with coefficients for salinity and temperature. The volume scattering coefficient of sea water is the sum of contributions from density fluctuations and concentration fluctuations and has been discussed in Zhang et al. (2009). Figure 1 shows the absorption and scattering coefficients for salinity S=20 PSU and temperature T=20°C.

### 2.3.2 Water constituents

The bio-optical model applies five IOPs. In addition to Chlorophyll pigment absorption, particle scattering and decaying organics absorption are each represented by a mixture of two different spectral coefficients (Doerffer et al., 2012). In this way, the natural variability of these IOPs can be retrieved along a gradient between two extremes, and assumptions regarding a specific spectral shape with spatial homogeneity are avoided. The spectral model for the detritus and CDOM absorption are derived from Prieur and Sathyendranath (1981). The 5 IOPs and and corresponding mathematical definitions, are namely:



– Absorption by phytoplankton (or chlorophyll) pigments (Figure 2):

$$\mathrm{apig}(\lambda) \,(\text{as measured by Doerffer}) \tag{2}$$

– Absorption by detritus (Figure 3):

$$\mathrm{ad}(\lambda) = \mathrm{ad}(443) \cdot e^{(-0.01(\lambda-443))} \tag{3}$$

– Absorption by CDOM (Figure 3):

$$\mathrm{ag}(\lambda) = \mathrm{ag}(443) \cdot e^{(-0.02(\lambda-443))} \tag{4}$$

– Particle scattering, white (large) particles (Figure 4):

$$\mathrm{bwhit}(\lambda) = \mathrm{bwhit}(550) \tag{5}$$

– Particle scattering, blue (small) particles (Figure 4):

$$\mathrm{bblue}(\lambda) = \mathrm{bblue}(550) \cdot e^{(-2(\lambda-550))} \tag{6}$$

And the aggregates thereof:

– Absorption by decaying organics (Figure 3):

$$\mathrm{adg}(\lambda) = \mathrm{ad}(\lambda) + \mathrm{ag}(\lambda) \tag{7}$$

– Total particle scattering (Figure 4):

$$\mathrm{bpart}(\lambda) = \mathrm{bwhit}(\lambda) + \mathrm{bblue}(\lambda) \tag{8}$$

The phase functions for the two scatterers are Fournier Forand phase functions with different back-scatter fractions. The back-scatter fractions are B=0.001 for large particles (bwhit) and B=0.1 for small particles (bblue). Fig. 5 shows a comparison of the two phase functions to the Petzold phase function (Petzold, 1972), which can be reproduced by a mixture of the two Fournier Forand phase functions.

Table 1 shows the resulting dimensions of the MOMO LUT. Table 2 shows relevant quantities that can be derived from the dimensions of the MOMO LUT.



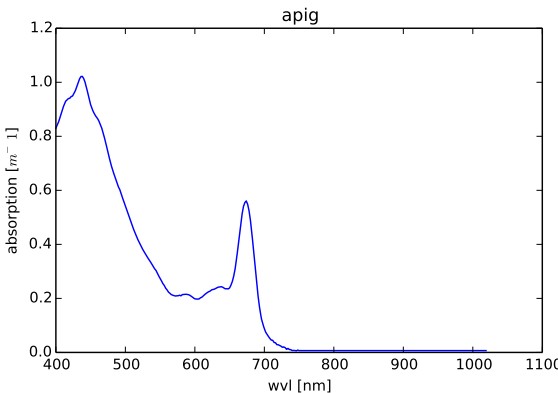

**Figure 2.** Spectral absorption of Chlorophyll-a-pigment as implemented in the RTM.

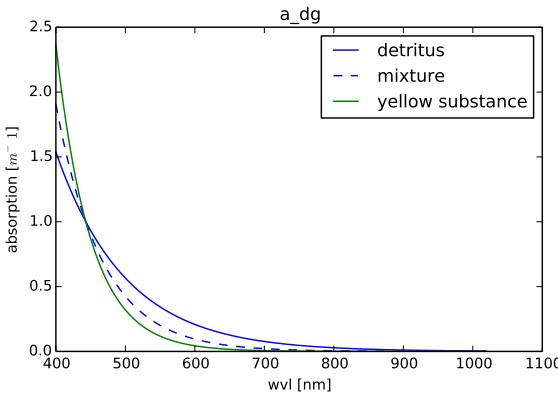

**Figure 3.** Spectral absorption of detritus and CDOM and a mixture of both as implemented in the RTM.

## 3 Calculation of the $R_{\mathrm{rs}}$ database

The $R_{\mathrm{rs}}$ is not a direct model output, but is derived from up- and downward radiances (L↑, L↓) and irradiances (E↑, E↓) just above water surface:

$$Rrs(\theta,\phi,\lambda) = L_w(\theta,\phi,\lambda)/E\downarrow(\lambda) \tag{9}$$

5   where the water-leaving radiance $L_w$ is calculated from

$$L_w(\theta,\phi,\lambda) = (L\uparrow(\theta,\phi,\lambda) - L_{black}(\theta,\phi,\lambda))/E\downarrow(\lambda) \tag{10}$$

and $L_{black}$ is L↑ from only the ocean surface. This is realised in the model, by implementing a very thin water body with a black surface below.





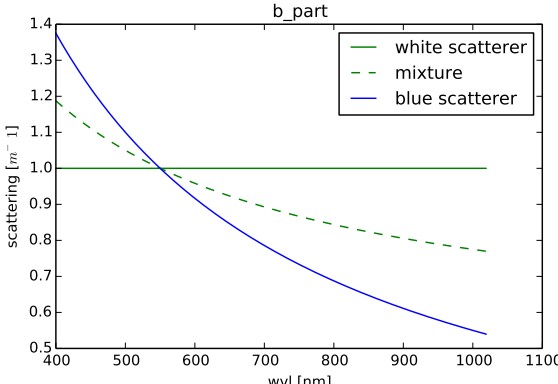

**Figure 4.** Spectral scattering of the white and blue scatterer and a mixture of both as implemented in the RTM.

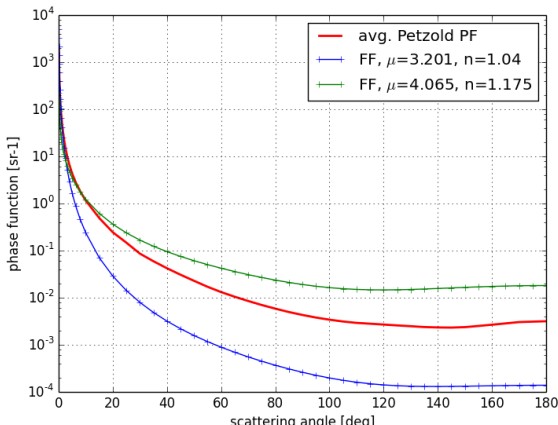

**Figure 5.** Comparison of Fournier Forand phase function for B=0.001 (blue) and B=0.1 (green) and Petzold phase function.

The IOP grid points are distributed on a logarithmic scale (due to the law of Lambert-Beer ) like follows (SZA ,VZA ,AZA as defined in Sect. 2.1.1):

- apig: 0.0005, 0.0016, 0.0055, 0.0184, 0.0612, 0.2039, 0.6786, 2.258, 7.5133, 25.0

- adg: 0.0, 0.0003, 0.0017, 0.0098, 0.0555, 0.3115, 1.7478, 9.8046, 55.0

- btot: 0.0, 6e-05, 0.00053, 0.00403, 0.02986, 0.22088, 1.63315, 12.07444, 89.27002, 660.0

- aratio: 0.0, 0.5, 1.0, where 0 means only ayel

- bratio: 0.0, 0.82499, 1.0, where 0 means only bblue





**Table 1.** Dimensions (IOPs) of the MOMO LUT

| IOPs | Notation | Units |
|---|---|---|
| Salinity | sal | $\mathrm{gkg^{-1}}$ |
| Absorption by phytoplankton @440nm | apig | $\mathrm{m^{-1}}$ |
| Absorption by detritus and CDOM @443nm | adg | $\mathrm{m^{-1}}$ |
| Total scattering coefficient @550nm | btot | $\mathrm{m^{-1}}$ |
| Ratios of IOPs | | |
| Ratio of detritus to CDOM absorption @443nm | aratio | - |
| Ratio of blue to white scattering @550nm | bratio | - |

**Table 2.** Quantities (IOPs) that can be derived from the dimensions of the MOMO LUT

| IOPs | Notation | Derivation | Units |
|---|---|---|---|
| Absorption by detritus @443nm | adet | $\mathrm{adg} \cdot \mathrm{aratio}$ | $\mathrm{m^{-1}}$ |
| Absorption by CDOM @443nm | ayel | $\mathrm{adg} \cdot (1 - \mathrm{aratio})$ | $\mathrm{m^{-1}}$ |
| Blue scattering coefficient @550nm | bblue | $\mathrm{btot} \cdot (1 - \mathrm{bratio})$ | $\mathrm{m^{-1}}$ |
| White scattering coefficient @550nm | bwhit | $\mathrm{btot} \cdot \mathrm{bratio}$ | $\mathrm{m^{-1}}$ |
| Total back-scattering coefficient @550nm | bbtot | $\mathrm{bwhit} \cdot 0.001 + \mathrm{bblue} \cdot 0.1$ | $\mathrm{m^{-1}}$ |
| Absorption by detritus and CDOM @$\lambda$nm | | $\mathrm{adg} \cdot e^{(-0.01(\lambda - 443))}$ | $\mathrm{m^{-1}}$ |
| Total scattering coefficient @$\lambda$nm | | $\mathrm{bwit} + \mathrm{bblue} \cdot e^{(-2(\lambda - 550))}$ | $\mathrm{m^{-1}}$ |

There frequency is chosen in order to reproduce the full functional relationship between $R_{\mathrm{rs}}$ and IOP. Therefore the interpolation to any possible IOP combination in between is expected to give reasonable results. Nevertheless, there are combinations of IOPs, which are not probable to occur in nature (see chapter 4). Since this co-variation is a function of season and region and therefore variable, we kept the LUT unconstrained. Figure 6 shows a subset of the database at wavelength=490nm, sza=43.64,

5  vza=12.55, aza=0.0, apig=0.2039, adg=0.0555, aratio=0.5, btot=0.02986, bratio=0.825, with each dimension varied over its grid points in a subplot.

## 4   Comparison to insitu measurements

The comparison is carried out on the basis of two different measures. On the basis of common colour ratios we checked if the MOMO LUT shows comparable sensitivities to the IOPs and spans the range of the insitu measurements. On the basis of

10  absolute values we compare the spectral shape of $R_{\mathrm{rs}}$ spectra.





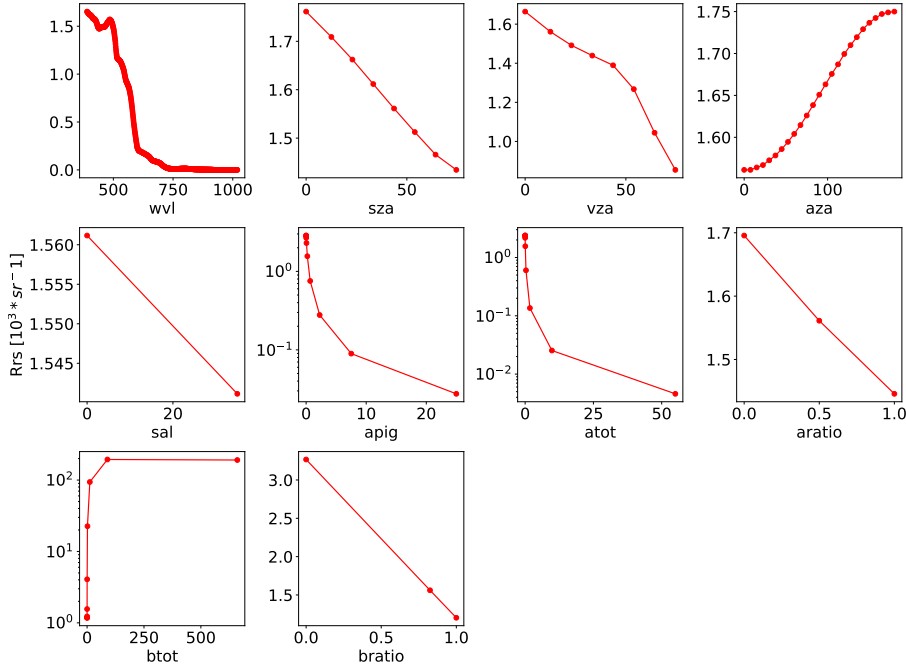

**Figure 6.** $R_{rs}$ simulated with the RTM MOMO at wavelength=490nm, sza=43.64, vza=12.55, aza=0.0, apig=0.2039, adg=0.0555, aratio=0.5, btot=0.02986, bratio=0.825, with each dimension varied over its grid points in a subplot.

1.) For the comparison on the basis of colour ratios we use a data set of insitu measurements which has been compiled for the C2X Project and is described in an Algorithm Theoretical Basis Document of this project (Hieronymi et al., 2017). Colour ratios are traditionally used for the retrieval of chlorophyll-a, but also for other IOPs. There are several different ratios used in literature (Nechad et al., 2015). For this comparison the ratios are chosen in order to be sensitive to the parameter of interest and to be available from insitu measurements. The insitu measurements are fully normalised, consequently the simulations are shown for Sun at zenith and viewing angle exactly perpendicular. While one specific IOP is given by the x-axis, the data is shown for all co-variations of the other IOPs in green. The functional relationship between the colour ratio and this specific IOP is then given by the sensitivity of the ratio to the IOP, but also represents the sensitivity of the co-varying IOPs to this ratio. In reality (represented by the insitu measurements) not all combinations of IOPs occur, but there is rather a specific relationship between them. For example, it is impossible to happen, that there is chlorophyll pigment absorption without scattering, because phytoplankton scatters itself. Also the appearance of chlorophyll pigment absorption without detritus is not probable, because detritus results from the natural decay of phytoplankton. In order to take those correlations into account, we masked some entries of the LUT by applying a very basic correlation and show the remaining LUT in yellow. The rule for the mask is done



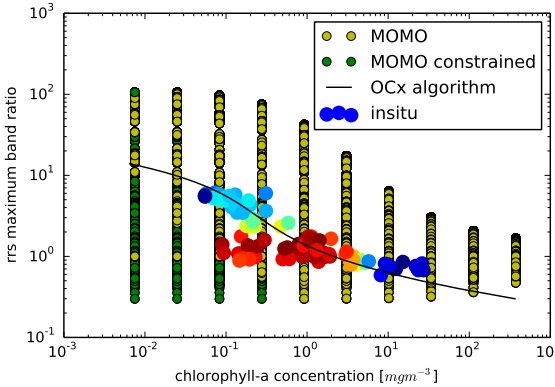

**Figure 7.** The maximum of $R_{\mathrm{rs}}(443)$, $R_{\mathrm{rs}}(490)$ and $R_{\mathrm{rs}}(510)$ divided to $R_{\mathrm{rs}}(555)$ as an indicator for chlorophyll-a from insitu measurements (blue: low, red: high number density) and from MOMO simulations (green) and masked (yellow).

on the basis of indices:

$$\mathrm{ibtot} \geq \mathrm{iapig} - 4$$
$$\mathrm{iadg} \geq \mathrm{iapig} - 4$$
$$\mathrm{ibtot} \geq \mathrm{iadg} - 4$$
$$\mathrm{iapig} \geq \mathrm{iadg} - 4$$
$$\mathrm{iadg} \geq \mathrm{ibtot} - 4$$
$$\mathrm{iapig} \geq \mathrm{ibtot} - 4 \tag{11}$$

As an indicator for chlorophyll-a we use the maximum of $R_{\mathrm{rs}}(443)$, $R_{\mathrm{rs}}(490)$ and $R_{\mathrm{rs}}(510)$ divided by $R_{\mathrm{rs}}(555)$, as shown in
Fig. 7. The black line shows the result of the OCx O'Reilly et al. (1998) algorithm. The simulations cover the range of band ratios for a certain chlorophyll-a concentration well and also the functional relationship is reproduced.

As an indicator for absorption of detritus and CDOM at 412.5nm we use $R_{\mathrm{rs}}(490)/R_{\mathrm{rs}}(670)$ as shown in Fig. 8. Again the simulations cover the insitu measurements and the filtered simulations reproduce the slope.

As an indicator for back-scatter at 510nm we use $R_{\mathrm{rs}}(490)/R_{\mathrm{rs}}(555)$ as shown in Fig. 9. Here, for the simulations we calculated
back-scatter as the sum of the back-scatter weighted scattering coefficient of the two scatterers. Again the simulations cover the insitu measurements. Regarding slope, the insitu measurements can be divided into two regimes, that are both covered by the simulations.

2.) In order to proof that the MOMO simulations are able to reproduce the spectral slope of $R_{\mathrm{rs}}$, we compare field measurements (without the issue of atmospheric correction) to the corresponding interpolated spectra from the LUT. For the validation
of our database we chose measurements from European lakes, because lake water has much more spectral variability than sea




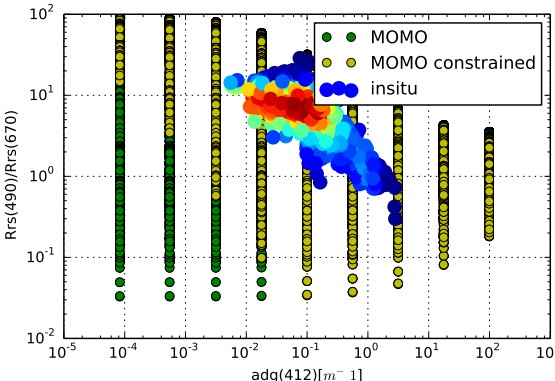

**Figure 8.** Absorption of detritus and CDOM at 412.5nm over $R_{\mathrm{rs}}(490)/R_{\mathrm{rs}}(670)$ from insitu measurements (blue: low, red: high number density) and from MOMO simulations (green) and masked (yellow).

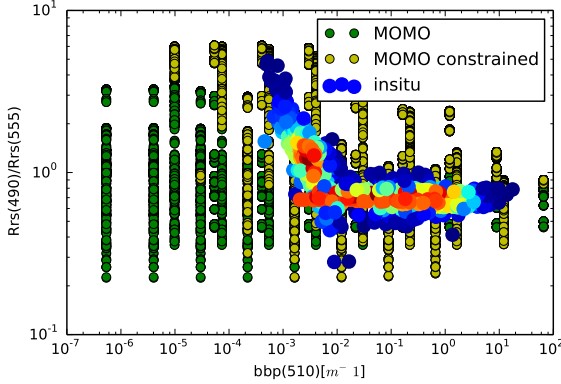

**Figure 9.** Back-scatter at 510nm over $R_{\mathrm{rs}}(490)/R_{\mathrm{rs}}(555)$ from insitu measurements (blue: low, red: high number density) and from our simulations (green) and masked (yellow).

water. The measured spectra were accompanied with measurements of chlorophyll concentration, CDOM absorption and TSM (total suspended matter) concentration. In order to get IOPs, we applied the following conversion factors:

$$\mathrm{TSM[mg/l]} = 1.73 \cdot \mathrm{btot} \tag{12}$$

$$5 \quad \mathrm{Chl[\mu g/l]} = 21 \cdot \mathrm{apig} \tag{13}$$

Table 3 shows the IOPs that are inferred from those conversion factors.

Furthermore we assumed aratio=0, which means only CDOM, but no Detritus absorption. bratio was varied in order to get the best fit, because we have no measurement about the size or the phase function of the particles available. Figure 10



**Table 3.** IOP's inferred from insitu measurements.

| IOP $[m^{-1}]$ | apig | adg | btot |
|---|---|---|---|
| Lake Säkylän Pyhäjärvi | 0.357 | 1.5 | 1.156 |
| Lake Lammin Pääjärvi | 0.164 | 9.63 | 1.098 |
| Lake Garda | 0.35 | 0.2 | 1.79 |

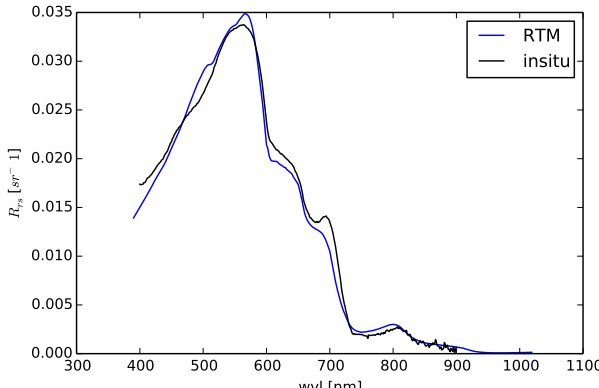

**Figure 10.** Comparison of an insitu measured $R_{\mathrm{rs}}$ spectrum at Lake Garda (Italy) with a corresponding $R_{\mathrm{rs}}$ spectrum from the MOMO simulations.

shows the measured and modelled $R_{\mathrm{rs}}$ from Lake Garda. Here as well the absolute value as the shape are reproduced well by the simulations. The comparison for Lake Lammin Pääjärvi (figure 11) shows good agreement in absolute values and shape, the simulated spectrum seems only a bit tilted compared to the measured $R_{\mathrm{rs}}$, which might be due to a slightly different chlorophyll absorption spectrum. Figure 12 shows the measured and modelled $R_{\mathrm{rs}}$ for Lake Säkylän Pyhäjärvi. Some features

5  are reproduced and also the absolute value of the peak at 550nm. The big difference in the red wavelength range might be due to stratification in the water column. In a nearby (4 km away) station a surface sample was taken which had clearly higher TSM and chlorophyll (with roughly the same TSM value from the composite sample from the surface to about 2 m depth). This example shows the limitations of the database due to the simplification of having only one water layer.

## 5   Conclusions

10  The remote sensing reflectance ($R_{\mathrm{rs}}$) contains the spectral colour information of a water body. Here $R_{\mathrm{rs}}$ are simulated using a RTM with a coupled atmosphere and ocean system by applying a bio-optical model based on IOPs. Those IOPs are varied over a global range. In the present article the configuration of the RTM is explained and discussed. On the basis of colour ratios as well the global range from insitu measurements as the sensitivity to the optical properties of the main water constituents could be reproduced. Only a subset of combinations of IOPs is required in order to cover the range of the measurements.



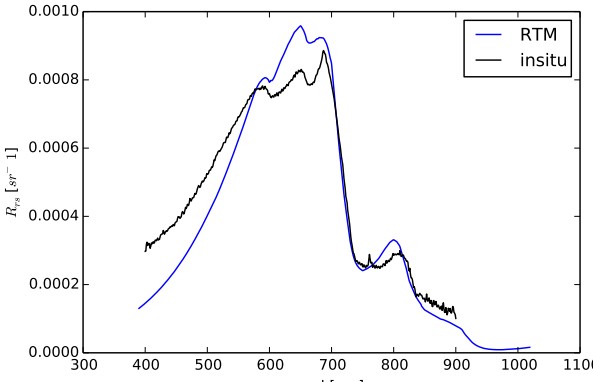

**Figure 11.** Comparison of an insitu measured $R_{\mathrm{rs}}$ spectrum at Lake Lammin Pääjärvi (Finland) with a corresponding $R_{\mathrm{rs}}$ spectrum from the MOMO simulations.

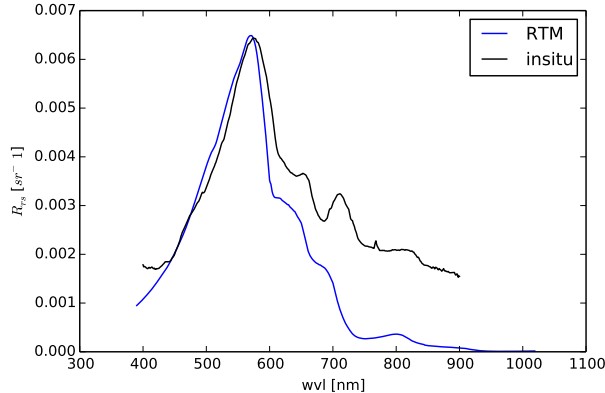

**Figure 12.** Comparison of an insitu measured $R_{\mathrm{rs}}$ spectrum at Lake Säkylän Pyhäjärvi (Finland) with a corresponding $R_{\mathrm{rs}}$ spectrum from the MOMO simulations.

Hyper-spectral measurements of $R_{\mathrm{rs}}$ at different European lakes agree well with spectra from the MOMO LUT in shape and absolute values. The simulations are suitable to be applied as a forward model for the retrieval of IOP's and water constituents from measured $R_{\mathrm{rs}}$s.

## 6 Code availability

5 The radiative transfer code MOMO is hosted at the institute of space sciences at Freie Universität Berlin and not publicly available.





## 7 Data availability

The database is publicly available from the World Data Center for Climate (https://doi.org/10.1594/WDCC/LUT_for_WDC_I, Kritten et al. (2017))

*Author contributions.* RP and JF were guiding the implementation of the bio-optical model into the RTM MOMO and the plausibility checks
5 of the results. CB was masterminding the definition of input optical properties of the water body for the simulations and participated in the scientific assessment of the spectra. TF participated in the scientific assessment of the spectra. SK delivered in-situ measurements used for validation and conceptional discussions.

*Competing interests.* The authors declare that they have no conflict of interest.

*Acknowledgements.* This work was founded through the BmWI/DLR project SIOCS. We thank Kerstin Stelzer and Daniel Odermatt for
10 fruitful discussions throughout the whole project. We thank Claudia Giardino and Mariano Bresciani for providing insitu measurements from Lake Garda, which were founded by the Glass project.



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
