# Peer review of "Water remote sensing reflectance from radiative transfer simulations on a global scale of Inherent Optical Properties"

_Earth System Science Data, 2018_

## Referee Comment (RC1) · Anonymous Referee #1 · 24 Apr 2018

In its present form, the manuscript is closer to an internal project report than to a manuscript. Writing is approximate, the introduction is 8-line long, many quantities are not defined, quality of illustration is poor, justification of the choice of parameters quasi absent etc...

Authors should go through published ESSD papers in order to get some feeling about what's expected in this journal.

---

## Referee Comment (RC2) · Anonymous Referee #2 · 11 May 2018

General comment

This manuscript presents a potentially useful dataset. Unfortunately it is partly unclear how it has been generated, several assumptions made are not explained, physical units are missing, and finally comparisons between measured and modelled IOPs and Rrs spectra are not convincing (Figures 7-9 and Figures 10-12, respectively). For these many reasons I do not recommend publication of this study at this stage.

Detailed comments

Page 1, line 18 Many databases but only 2 references?

Page 2, equation 1 Please provide physical units for each parameter

Page 2, 2.2 Atmosphere Why only marine aerosols? Why not considering also continental aerosols when dealing with coastal and inland waters?

Page 3, Figure 1 Indicate the sources/references for this data

Page 3, lines 5-8 Unit of backscattering? Is it realistic to consider such high (400 1/m?) backscattering?

Page 3, line 18 Spectral model of Prieur and Sathyendranath (1981) is quite old, there is no more recent model?

Page 4, equations 2-8 Provide units for each parameter apig as measured by Doerffer: reference?? Equation 5: why white large particles? Any reference to justify this? Equation 6: should be a power-law function, not exponential, or provide reference for this Equation 8: why mixing blue and white particles to get a realistic particulate scattering? Why not using Petzold and realistic variations arount it? Please justify. "The backscatter fractions are B=0.001 for large particles (bwhit) and B=0.1 for small particles (bblue)"?? why such low and high fractions?? Can you ustify this based on literature? Yu must provide references here.

Page 5, Figure 3 What about non-negligible light absorption by detritus in the 700-110 nm spectral region? See for example Estapa et al. (2012) Role of iron and organic carbon in mass‐specific light absorption by particulate matter from Louisiana coastal waters, Limonl. Oceanogr., https://doi.org/10.4319/lo.2012.57.1.0097.

Page 6, apig to bratio Totally unclear if you do not specify the wavelengths And each parameter must be defined (btot, aratio)

Page 7, line 3 'Nevertheless, there are combinations of IOPs, which are not probable to occur in nature ' True, unrealistic cases must be filtered.

Page 8, Figure 6 Provide units in the legend

Figures 7-9: I am not convinced by these comparisons between simulated and measured datasets, I think many cases simulated are not realistic

Page 9 'ibtot > iapig−4 iadg > iapig−4 ibtot > iadg−4 iapig > iadg−4 iadg > ibtot−4 iapig > ibtot−4' Totally unclear

'we use Rrs(490)/Rrs(670) . . . we use Rrs(490)/Rrs(555)' Why using these specific ratios? Please justify based n references

Page 10 Equations 12-13 Why such values? You must provide references

Pages 11-12, Figures 10-12 Why only 3 comparisons between modeled and measured Rrs spectra are presented? How representative are these 3 cases? Comparisons in Figures 11 and 12 are not convincing as large differences can be observed between modeled and measured Rrs and are not explained.